# Targeted Vaccine Allocation Could Increase the COVID-19 Vaccine Benefits Amidst Its Lack of Availability: A Mathematical Modeling Study in Indonesia

**DOI:** 10.3390/vaccines9050462

**Published:** 2021-05-06

**Authors:** Ahmad Fuady, Nuning Nuraini, Kamal K. Sukandar, Bony W. Lestari

**Affiliations:** 1Department of Community Medicine, Faculty of Medicine, Universitas Indonesia, Jakarta 10310, Indonesia; 2Department of Public Health, Erasmus MC University Medical Center Rotterdam, 3015 GD Rotterdam, The Netherlands; 3Department of Mathematics, Institut Teknologi Bandung, Bandung 40132, Indonesia; nuning@math.itb.ac.id (N.N.); kamalkhairudin@students.itb.ac.id (K.K.S.); 4Epidemiology Group of COVID-19 Task Force for West Java, Bandung 40171, Indonesia; bony.wiem@unpad.ac.id; 5Department of Public Health, Faculty of Medicine, Universitas Padjadjaran, Bandung 40161, Indonesia; 6Department of Internal Medicine, Radboud University Medical Center, 6525 GA Nijmegen, The Netherlands

**Keywords:** COVID-19, vaccine, strategy, modeling, low- and middle-income countries

## Abstract

With a limited number of vaccines and healthcare capacity shortages, particularly in low- and middle-income countries, vaccination programs should seek the most efficient strategy to reduce the negative impact of the COVID-19 pandemics. This study aims at assessing several scenarios of delivering the vaccine to people in Indonesia. We develop a model for several scenarios of delivering vaccines: without vaccination, fair distribution, and targeted distribution to five and eight districts with the highest COVID-19 incidence in West Java, one of the most COVID-19-affected regions in Indonesia. We calculate the needs of vaccines and healthcare staff for the program, then simulate the model for the initial 4-month and one-year scenarios. A one-year vaccination program would require 232,000 inoculations per day by 4833 vaccinators. Targeted vaccine allocation based on the burden of COVID-19 cases could benefit the COVID-19 vaccination program by lowering at least 5000 active cases. The benefits would increase by improving the number of vaccines and healthcare staff. Amidst lacking available vaccines, targeted vaccine allocation based on the burden of COVID-19 cases could increase the benefit of the COVID-19 vaccination program but still requires progressive efforts to improve healthcare capacity and vaccine availability for optimal protection for people.

## 1. Introduction

The unprecedented pandemic caused by coronavirus disease 2019 (COVID-19) leads to a global race to find the most effective vaccine and deliver it most efficiently [1,2]. However, when the COVID-19 vaccination aims at achieving global herd immunity, an estimated 5.5 billion out of a total of 7.8 billion world population would demand 11 billion doses of COVID-19 vaccines [3]. With a limited capacity of an estimated 6.2 billion doses per year and the remaining needs for other vaccines, such as those for rubella, polio, and tuberculosis at the same time, the availability of the vaccine would not be sufficient to supply the whole world [4,5].

Indonesia, with its 270 million people, has the same problem regarding vaccination supply side. With a target to inoculate 181 million people, Indonesia requires at least 426 million vaccine doses [6]. Although Indonesia had settled an agreement with Sinovac, a vaccine produced by a China-based biopharmaceutical company [7] for most of its population, the successful COVID-19 vaccination program requires a robust logistic chain, systematic distribution, and a good level of people’s acceptance toward COVID-19 vaccination [8,9]. These problems, along with the base critical issue of vaccine shortage, require a breakthrough strategy to maximize the benefits for people.

The critical challenge is to seek the best scenario to deliver the vaccine to the people. There are two main options: fair and comprehensive distribution to as many people or targeted distribution to a much more impacted region. We aimed to answer this question by developing four main scenarios in West Java, one of the most COVID-19-affected provinces in Indonesia.

## 2. Materials and Methods

### 2.1. Context

In February 2021, confirmed COVID-19 infection in Indonesia had surpassed 1 million cases with an estimated 27,000 deaths [10]. The actual numbers of cases and deaths are potentially higher than the recorded figures. West Java is a province with the second-highest number of COVID-19 confirmed cases after the Indonesian capital, Jakarta. However, different from the urban, small city context of Jakarta, West Java has a vast population of 49.94 million people and a wide geographical area that comprises 27 districts/cities varying from urban to rural.

The Indonesian government has prioritized healthcare workers, public sector employees, and laypersons in adult groups (defined as 18–59 years old) and lately included the older population into the prioritized group with caution [11]. Although the government has confirmed 300 million doses of vaccine from Sinovac, Pfizer, AstraZeneca, and other vaccines from the CEPI initiative, the initially available doses in the first six months will not exceed 50 million doses. The limited vaccines should also be distributed to 34 provinces in Indonesia, leaving West Java with a much more limited number of vaccines for inoculation.

### 2.2. Model of Disease Transmission

We used a mathematical model representing the spread of COVID-19 using five compartments, i.e., susceptible, infected, quarantined, recovered, and deaths, called as SIQRD model [12] (Figure 1). This model is an extension of the basic mathematical modeling that has been commonly used for other virus-induced illnesses, such as influenza [13,14]. Compartment Q is an additional compartment in which we assumed that persons would not become infectious because of quarantine. Their mobility during quarantine is restricted and supervised by medical staff.

Assuming that vaccine would give acquired immunity and protection, we formulated v(t) as a rate of vaccinated persons who developed acquired immunity, involving the estimation of vaccine efficacy. However, persons in compartment R could be transferred back to S due to the chance of reinfection. The vaccine might not give the perfect protection from infection due to the mismatch problem of the type of virus being used for the vaccine. Thus, instead of defining it as the total number of recovered persons, we defined R as the total number of immune people due to recovery and reinfection.

We formulated a system of differential equations. Since Indonesia has initially prioritized only the young adult population for inoculation, we use the non-age structure model. As we use S(t) to denote the susceptible population at time-t, we defined the model as below.
(1)$$\frac{dS\left(t\right)}{dt}=\pi \hat{N}\left(t\right)+\zeta R\left(t\right)-\left(\beta \frac{I\left(t\right)}{N}+\eta v\left(t\right)+\mu \right)S\left(t\right)$$
(2)$$\frac{dI\left(t\right)}{dt}=\beta \frac{S\left(t\right)I\left(t\right)}{N}-\left(q+\mu \right)I\left(t\right)$$
(3)$$\frac{dQ\left(t\right)}{dt}=qI\left(t\right)-\left(\gamma +\delta +\mu \right)Q\left(t\right)$$
(4)$$\frac{dR\left(t\right)}{dt}=\gamma Q\left(t\right)+\eta v\left(t\right)S\left(t\right)-\left(\zeta +\mu \right)R\left(t\right)$$
(5)$$\frac{dD\left(t\right)}{dt}=\delta Q\left(t\right)$$

$\hat{N}\left(t\right)$ was defined as the total living person in a population, given by $\hat{N}\left(t\right)=N-D\left(t\right).$ We also assumed that π=μ since, during the COVID-19 pandemic, we could not find the factual numbers of birth and death rate with a very dynamic situation. Instead of using the real numbers, we decided to ignore them and assume that the recruitment rate equals the natural death rate in a short period of observation time. This assumption implied that the number of people is constant during the time horizon. Details of variables and parameters are given in Table A1.

The vaccination rate, v(t), is a complex variable and difficult to be estimated since it can vary over time and is affected by many variables. To bypass the computational problems caused by these properties, we made the simplifying assumption that v(t) is constant over time. By this assumption, the inoculation rate should also be kept consistent. Clearly from Equations (1)–(5) and assuming all parameters to be time-independent, the dynamics of S(t), I(t), Q(t), R(t), and D(t) depend on the chosen value of v(t)=v.

In this modeling, we developed several scenarios based on (a) time-period and (b) targeted district distribution. Because of limited access to specific data, the two aspects’ analysis will be narrowed specifically to West Java, equipped with 27 cities/districts in total.

### 2.3. Time-Period Scenarios

The Ministry of Health, Republic of Indonesia, targeted 67 million inoculations to 33.5 million people in West Java. We developed two main scenarios based on the program implementation’s targeted period, which would be accomplished in (a) six months and (b) twelve months. We then calculated the number of health care staff and vaccines dosage per day needed for these two scenarios with the assumption of 10 min per person per inoculation.

### 2.4. Targeted Allocation Scenarios

However, by the end of 2020, the available vaccine distributed to West Java in the first four months would be only 97,080 dosages. This is a severe vaccine limitation for inoculating vaccine to 33.5 million people. West Java has 27 cities/districts with the various burden of COVID-19. We can distribute the limited number of vaccines fairly across the districts or focus on several districts in which the COVID-19 has highly impacted. To decide the most effective distribution strategy, we developed several scenarios, which were (a) baseline without vaccination, (b) fair distribution to all 27 districts, (c) targeted distribution to five districts with the highest number of COVID-19 cases, and (d) targeted distribution to eight districts with the highest COVID-19 cases. We also assumed that the inoculation would be strictly delivered based on residence, and mobility between districts would not be allowed.

## 3. Results

### 3.1. Time-Period Scenarios

The initial number of healthcare staff that are available for the COVID-19 vaccination program was 1094 persons. Those staff will be divided into three main tasks: registration and screening, vaccinator, and observation. With an equal ratio between the three tasks, the number of available vaccinators would be 365 persons. Suppose the average time of vaccination is 10 min per person, one vaccinator could inoculate 48 persons per day, and the total average inoculation per day would be 17,504 (Table 1).

The current healthcare staff is severely lacking. If the vaccination program should be accomplished in 12 months, it requires 232,000 inoculations per day and needs 4833 vaccinators to do the task. The number would double up to 9667 vaccinators if the target period is shortened to a six-month program. Otherwise, the duration per person inoculation should be reduced to 5 min per person, then it can halve the required number of staff for the program.

### 3.2. Targeted Allocation Scenarios

The mean effective reproduction number (R_E_) values for 27 districts in West Java was 1.38, assuming no vaccination program with the current mobility restriction estimate. The mean R_E_ values for five districts and eight districts with the highest COVID-19 cases were 1.26 and 1.25, respectively. With this R_E_, the model simulations showed that the number of infections would increase until September 2021, followed by a decreasing infection rate.

The modeling simulation also showed that the COVID-19 vaccination could reduce the number of COVID-19 cases significantly (Figure 2). If the vaccine can be distributed optimally within 12 months, the cases could be reduced much earlier until less than 30,000 cases per month. If the vaccine could be delivered in a much faster way, six months, the cases can be controlled after a four-month inoculation program.

These two scenarios were very optimistic. Given the limited availability of the vaccine, we simulated several scenarios by distributing the vaccines according to the COVID-19 incidence in each district. By 23 January 2021, it has been confirmed that several cities/districts with the highest number of infections were Depok City, Bekasi City, Karawang District, Bekasi District, Bandung District, Tasikmalaya District, Bandung City, and Bogor City. The details of COVID-19 cases in 27 cities/districts are provided in Table A2.

Since the currently available number of vaccines was significantly smaller than the needs (97,080 vs. 22,000,000 dosages) for the first four months, the inoculation would not reduce the total number of active cases significantly (Figure 3a). However, distributing the vaccine by targeting allocation to the most impacted cities/district is the most worth-trying scenario that could decrease 5000 cases.

Assuming that the government could not improve the vaccine capacity within the first year, there would be an estimated 300,000 inoculation in one year until December 2021. By this assumption, after six months, we could see the effects of targeted vaccine allocation (Figure 3b). Targeted vaccine allocation to the five and eight cities/districts with the highest COVID-19 incidence would deliver more benefits than distributing the vaccine equally to all districts. At the peak of the COVID-19 transmission curve, targeted vaccine allocation could reduce the new cases, ranging from 5000 to 8000 cases. However, after reaching the curve peak and the decreasing infection rate, there would be no significant differences between scenarios.

The effect would be more significant if the number of vaccines could be increased as targeted, 33 million dosages in one year. By this assumption, the targeted allocation would not be effective, while equal distribution between districts would provide the most benefit to halting the infection rate (Figure 3c).

## 4. Discussion

The findings suggested that, in case of lacking available vaccines, targeted vaccine allocation based on the burden of COVID-19 cases could increase the benefit of the COVID-19 vaccination program. The targeted allocation to the district with the highest COVID-19 incidence could lower the total number of COVID-19 active cases as many as 5000 cases compared with the cases in fair distribution. The reduction of the cases could be much higher when the transmission curve reaches its peak. However, if the vaccines are severely lacking, the impact of targeted vaccine allocation would be modest. The combination of targeted allocation and increasing vaccines’ availability is the most crucial strategy to halt the COVID-19 transmission.

The vaccination program in a shorter period will give more benefit in infection control. However, it may be impossible given the limited capacity of healthcare and the availability of vaccines. Extraordinary efforts are required to achieve the target of hundred-thousand inoculation per day and increase the number of healthcare staff up to more than seven times the current number within a limited, tight time. One plausible strategy is the involvement of community health workers who are familiar with child vaccination or tuberculosis program [15]. In low- and middle-income countries, the problem is beyond the healthcare staff availability. It also includes inadequate infrastructure for vaccination [16]. Provision of million dosages per day would require sufficient storage and supervision in the whole cold chain to prevent vaccine wastage.

The model we developed in this study is different from most models developed in many countries that prioritize vaccination for the elderly, such as simulated scenarios for India [2], United States, Germany, and South Korea [17]. Those models showed that prioritizing COVID-19 vaccine allocation for older populations (i.e., >60 years) resulted in the most significant relative reduction in deaths and could maximize years of remaining life expectancy. Instead, the model we developed in this study was based on the Indonesian government’s initial prioritization strategy at the beginning of 2021, taking into account that Sinovac was the only vaccine product available when the policy was issued. This strategy prioritized populations aged 19–59 years old instead of the elderly and people with comorbidities. However, the simulation showed that this prioritization strategy could still have a positive effect on infection control. Vaccination with this approach could reduce the number of cases, although these reductions in the scenarios are marginal. 

Nevertheless, although the vaccination strategy may reduce the absolute number of infections, it does not necessarily reflect the effects on the overall COVID-19 control, including the reduction of mortality rate. Mortality due to COVID-19 is higher in the elderly population [10]. Given the higher mortality in the elderly group, the COVID-19 vaccine should be equally allocated, and the elderly should be prioritized [1,18]. The high-risk group prioritization was also adopted by many other institutions, such as the National Institutes of Health (NIH), the Centers for Disease Control and Prevention (CDC), and the National Academies [19]. 

Given that elderly was not the priority in this context, and infection to this group has a higher risk for hospitalization and intensive care, the healthcare burden could remain higher. Therefore, vaccination to health care workers and social workers should be prioritized because of the high risk they are facing in healthcare setting [20]. Indonesia, like other LMICs, also has to deal with healthcare workforce shortage [21], and it highlights that vaccinating this substantial population should be placed as a high priority. Aside from the vaccination, they should also be equipped with an adequate supply of personal protective equipment and psychosocial support, such as sufficient time for resting, appropriate temporary housing nearby healthcare facilities, mental aid support when needed, and improvement of decision system to work removal and return to work [22,23].

Models developed by Olivares and Stafetti (2021) showed that even a mass vaccination with one million inoculations per week during six months would only produce 15–18% of acquired immunity. That would add the proportion of immunized individuals to 46–59% of the total population that is still not sufficient for reaching herd immunity [24]. With limited vaccine availability and health care capacity, optimal protection could not be reached. It is also much farther from achieving herd immunity. The targeted allocation may reduce the number of COVID-19 cases but is less likely to halt to a significantly much lower number. Aside from increasing the number of available vaccines, a targeted allocation strategy to achieve optimum protection also requires robust data management to identify the most impacted area. It is critical to help the government develop robust planning of the national and local vaccine supply and deployment.

Among the problems, this modeling suggested that to achieve maximum protection for people by providing adequate vaccine supply means that government should increase the budget for vaccination programs. While the government may have to deal with funding shortages, support from the private sector is needed [25]. However, collaboration with the private sector should not harm equal and fair distribution across the population. A public-private partnership should not neglect distributive justice when the potential of purchasing vaccines through private sectors may arise. Vaccine stewardship should also be governed in a multivalue ethical framework [26]. Before pandemic, health and socioeconomic disparities existed, and failure to promote good vaccination governance and stewardship will only exacerbate the disparities.

The vaccination program also requires clear communication and message to the public, including explaining their plans and evaluation [27]. On the other hand, the public should encourage transparency during the vaccination program to increase public trust and acceptance of vaccines [28].

This modeling based on West Java data and therefore has a limitation on its generalizability. Findings in this study should be carefully interpreted when translating to other contexts. This study also did not include non-pharmaceutical intervention and the dynamic situation. Regarding a developed understanding of COVID-19 immunity dynamic and uncertainties of vaccine delivery timelines and duration of developed immunity from natural infection, this study did not include all these parameters into the model due to unavailable reliable data. Although this model helps compare allocation strategies, it does not necessarily provide the exact estimates of cumulative COVID-19 cases.

## 5. Conclusions

Although the vaccines are ready for distribution, there remain challenges of fair and optimal allocation to reduce and halt the infection rate optimally. In case of lacking available vaccines, targeted vaccine allocation based on the burden of COVID-19 cases could increase the benefit of the COVID-19 vaccination program. However, this strategy still requires progressive efforts to improve healthcare capacity and vaccine availability to achieve optimal protection for people.

## Figures and Tables

**Figure 1 vaccines-09-00462-f001:**
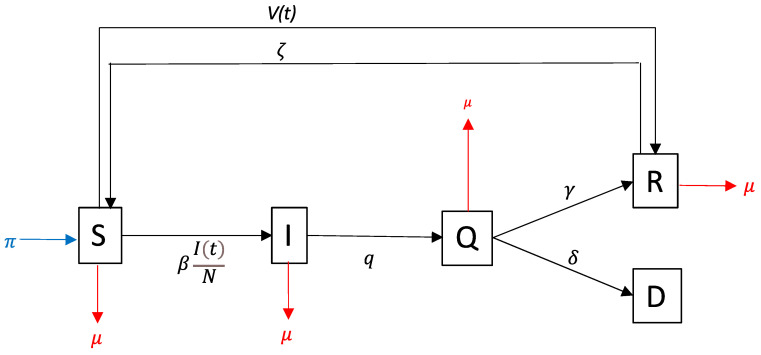
The SIQRD model applied in this study. S, susceptible; I, infected; Q, quarantined; R, recovery; D, death. Black arrows represent the logic flow of infection using SQRD model. Blue arrows represent natural recruitment (π). Red arrows represent the death rate (μ). The detailed description of variables is shown in Appendix A.

**Figure 2 vaccines-09-00462-f002:**
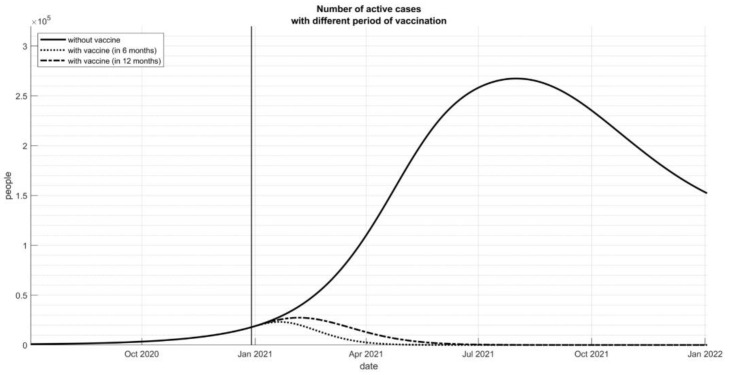
Number of active cases with different period of vaccination.

**Figure 3 vaccines-09-00462-f003:**
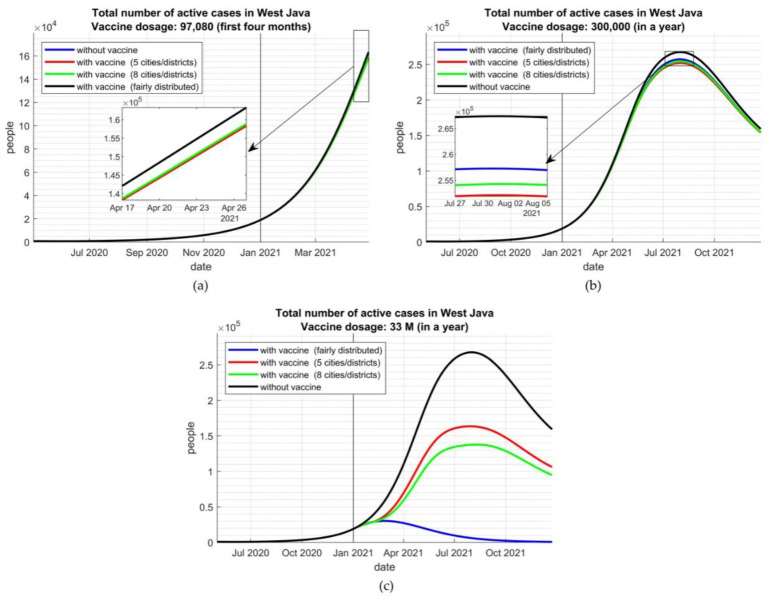
The number of active cases with several scenarios of vaccine distribution given (**a**) the severely lacking vaccines—97,080 dosages in the first four months, (**b**) 300,000 dosages in one year, and (**c**) the optimum available vaccines—33,000,000 dosages in one year.

**Table 1 vaccines-09-00462-t001:** Estimation of required numbers of healthcare staff, vaccinators, and vaccine dosages per day for 6- and 12-month periods.

Variables	Availability	Targeted Time-Period			
		6 Months		12 Months	
# of targeted inoculation	33,500,000				
# of vaccine dosage ^a^	67,000,000				
# of healthcare staff	1094	29,000	14,500	14,500	7250
# of vaccinator	365	9667	4833	4833	2417
Duration (minute/person)	10	10	5	10	5
Inoculation per day ^b^	48	48	96	48	96
# of vaccine per day	17,504	464,000	464,000	232,000	232,000

#, number; ^a^ two-dosage per person; ^b^ estimated by 8-working hour per day.

## Data Availability

The data presented in this study are available on request from the corresponding author.

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
