# Peer review of "Targeted Vaccine Allocation Could Increase the COVID-19 Vaccine Benefits Amidst Its Lack of Availability: A Mathematical Modeling Study in Indonesia"

_vaccines, 2021, doi:10.3390/vaccines9050462_

Round 1
Reviewer 1 Report
The authors of this interesting paper address the problem of how to
deal with the administration of vaccines when their availability is limited.
They focus on Indonesia, but the results of their observation are of great importance
for the whole world, as they state in the introduction of their manuscript.
They adopt a dynamical epidemic model of six coupled equations adapted to the vaccination issue and using this model they address the hard problem of administrating vaccines in Indonesia, country that should inoculate vaccine to 33.5 million people. Following the directions of the Indonesia government they adopt the targeted prescription that does not pay attention to age but to the nodes that when removed from the network fragment it into
small clusters. This is a bright choice that the authors try to improve paying attention
to health and social workers. This has the effect of saving lives, but not enough to fight
the severity of Covid-19 pandemic in Indonesia. They conclude their investigation stating that to reduce the high-mortality rate, attention should also be paid to age.
This is an interesting investigation deserving publication. I limit myself to notice that
some improvement to the English of the text should be done.
Line 110 Probably the sentence “However, in this mode we simplified the assumption that ?(?) was constant over time” must be turned into “To bypass the computational problems caused by these properties we made the
simplifying assumption that n is constant”.
Line 130. Probably they should reply “It was severely lacks compared to the need for inoculating vaccine to 33.5 million people.” With “This is a severe vaccine limitation for inoculating vaccine to 33.5 million people.”
Author Response
Response to Reviewer 1
Point 1. Line 110 Probably the sentence “However, in this mode we simplified the assumption that ?(?) was constant over time” must be turned into “To bypass the computational problems caused by these properties we made the simplifying assumption that n is constant”.
Response: First, we thank you for your comment and appreciation to our manuscript. Thank you for your suggestion. We revised the sentence accordingly (lines 112-114).
Point 2. Line 130. Probably they should reply “It was severely lacks compared to the need for inoculating vaccine to 33.5 million people.” With “This is a severe vaccine limitation for inoculating vaccine to 33.5 million people.
Response: Thank you for your suggestions. We revised this section accordingly (lines 133-134)
We hope that the revised manuscript has addressed all comments and suggestions. We also hope that readers can easily understand the paper, and that this paper will provide insightful information on the strategy for COVID-19 vaccination delivery especially in low-middle income countries. Thank you.
Kindly regards,
Ahmad Fuady
Corresponding author

Reviewer 2 Report
In this study, the authors developed a model for several scenarios of delivering vaccines: without vaccination, fair distribution, and targeted distribution to five and eight most impacted COVID-19 districts in West Java, one of the most COVID-19-affected regions in Indonesia.
The paper can be accepted for the publication after some major revisions.
Comments:
- Use the present time in the abstract. For example, change "We developed" to "We develop".
- In the introduction section, change "coronavirus disease 19" to "coronavirus disease 2019".
- The last equation in your model is not correct. This should be corrected by ?'(?)=?Q(?). Also, check Figure 1.
- Why you assume that ?=?? This should be biologically justified.
- All the numerical simulations should be checked after corrected the equation (5).
- Compare your results established in this study with other existing in
the literature. - There are some typos. The authors should carefully read the manuscript.
- Include and comment in the manuscript a recent and related works:
[1] Dynamics of SARS-CoV-2 infection model with two modes of transmission and immune response, MBE 17(5) (2020) 5326–5340.
[2] A delayed SIR epidemic model with a general incidence rate,
Electron. J. Qual. Theory Differ. Equ. (2013) 1-9.
[3] Epidemiological models in high school mathematics education,
Commun. Math. Biol. Neurosci. (2020).
Author Response
Response to Reviewer: 2
Point 1. Use the present time in the abstract. For example, change "We developed" to "We develop".
Response: Thank you for your suggestion. We revised the abstract and used present time.
Point 2. In the introduction section, change "coronavirus disease 19" to "coronavirus disease 2019".
Response: We revised the words accordingly (Line 34).
Point 3. The last equation in your model is not correct. This should be corrected by ?'(?)=?Q(?). Also, check Figure 1.
Response: Thank you for your correction. This is a typographical error, and we revised it accordingly (Line 102). We also checked Figure 1 and there is no typographical error in the Figure.
Point 4. Why you assume that ?=?? This should be biologically justified.
Response: We realize that, in biological sense, the natural birth rate is not equal the natural death rate. However, in the pandemic situation, we could not find the factual numbers of birth and death rate with a very dynamic situation. Instead of using the real numbers, we decided to ignore them and assume that the recruitment rate is equal the natural death rate because we observe the phenomena in a short period of time. This assumption implies the constant number of people in population over the short period of time. We also acknowledge that this assumption is a limitation of the study that should be interpreted carefully. We explained it on lines 105-108.
Point 5. All the numerical simulations should be checked after corrected the equation (5).
Response: We confirm that the error in Equation (5) is only a typographical error. It does not affect the numerical simulation that we already run.
Point 6. Compare your results established in this study with other existing in the literature.
Response: We added comparisons with other studies in Discussion section. Since our simulation was based on the Indonesian government's initial prioritization strategy that prioritized young adult populations (aged 19-59 year) instead of elderly and people with comorbidities, there are very few literatures discussing the same approach. We explained it on lines 227-235. There has been lacked of literatures discussing the severely lack of vaccine availability and healthcare capacity, particularly in low- and middle-income countries. We then compare this situation with the ideal mass vaccination program (Lines 256-259)
Point 7. There are some typos. The authors should carefully read the manuscript.
Response: Thank you for your comments. We read thorough the manuscript and fixed all typos.
Point 8. Include and comment in the manuscript a recent and related works:
[1] Dynamics of SARS-CoV-2 infection model with two modes of transmission and immune response, MBE 17(5) (2020) 5326–5340.
[2] A delayed SIR epidemic model with a general incidence rate, Electron. J. Qual. Theory Differ. Equ. (2013) 1-9.
[3] Epidemiological models in high school mathematics education, Commun. Math. Biol. Neurosci. (2020).
Response: Thank you for your suggestion. The papers are very interesting in mathematical modelling. However, we inserted only the second paper (A delayed SIR epidemic model with a general incidence rate) in Method section since it discussed about S-I-R model (Line 77). The remaining papers are not correlated directly with model that we developed in this study. The first paper discussed a within-host model describing the interaction between SARS-CoV-2, host pulmonary epithelial cells and cytotoxic T lymphocyte (CTL) cells, while our paper run a model in a broader context: community. The third paper discussed epidemiological models in high school education which is not related to vaccine strategy.
We hope that the revised manuscript has addressed all comments and suggestions. We also hope that readers can easily understand the paper, and that this paper will provide insightful information on the strategy for COVID-19 vaccination delivery especially in low-middle income countries. Thank you.
Kindly regards,
Ahmad Fuady
Corresponding author

Round 2
Reviewer 2 Report
Now, the paper is well written, but it remains to do the following:
- In figure 1, change 'h' to 'q'.
- The citation of some references in the text is not correct. This should be checked.
Author Response
Response to Reviewer: 2
Point 1. In figure 1, change 'h' to 'q'.
Response: We changed ‘h’ to ‘q’ as now displayed in Figure 1.
Point 2. The citation of some references in the text is not correct. This should be checked.
Response: We revised the references and correct the citation in the text accordingly.
We hope that the revised manuscript has addressed all comments and suggestions. We also hope that readers can easily understand the paper, and that this paper will provide insightful information on the strategy for COVID-19 vaccination delivery especially in low-middle income countries. Thank you.
Kindly regards,
Ahmad Fuady
Corresponding author
